# Causes of Moderate and Severe Anaemia in a High-HIV and TB-Prevalent Adult Population in the Eastern Cape Province, South Africa

**DOI:** 10.3390/ijerph20043584

**Published:** 2023-02-17

**Authors:** Don O’Mahony, Sikhumbuzo A. Mabunda, Mbulelo Mntonintshi, Joshua Iruedo, Ramprakash Kaswa, Ernesto Blanco-Blanco, Basil Ogunsanwo, Kakia Anne Faith Namugenyi, Sandeep Vasaikar, Parimalaranie Yogeswaran

**Affiliations:** 1Department of Family Medicine and Rural Health, Walter Sisulu University, Mthatha 5117, South Africa; 2School of Population Health, University of New South Wales, Sydney 2052, Australia; 3George Institute for Global Health, University of New South Wales, Sydney 2042, Australia; 4Department of Public Health, Walter Sisulu University, Mthatha 5117, South Africa; 5Department of Laboratory Medicine and Pathology, Walter Sisulu University, Mthatha 5100, South Africa; 6Department of Surgery, Walter Sisulu University, Mthatha 5117, South Africa; 7Department of Microbiology, Walter Sisulu University, Mthatha 5117, South Africa

**Keywords:** anaemia causes, HIV, tuberculosis, nutritional deficiencies, anaemia of inflammation, primary care

## Abstract

Background: Anaemia affects one in four adults in South Africa, with a higher prevalence in persons with HIV and tuberculosis. The aim of this study is to characterise the causes of anaemia in primary care and a district hospital setting. Methods: A cross-sectional study design investigated a purposive sample of adult males and non-pregnant females at two community health centres and a hospital casualty and outpatients. Fingerpick blood haemoglobin was measured with HemoCueHb201+. Those with moderate and severe anaemia underwent clinical examination and laboratory tests. Results: Of 1327 patients screened, median age was 48 years, and 63.5% were female. Of 471 (35.5%) with moderate and severe anaemia on HemoCue, 55.2% had HIV, 16.6% tuberculosis, 5.9% chronic kidney disease, 2.6% cancer, and 1.3% heart failure. Laboratory testing confirmed 227 (48.2%) with moderate and 111 (23.6%) with severe anaemia, of whom 72.3% had anaemia of inflammation, 26.5% iron-deficiency anaemia, 6.1% folate deficiency, and 2.5% vitamin B12 deficiency. Overall, 57.5% had two or more causes of anaemia. Multivariate modelling showed that patients with severe anaemia were three times more likely to have tuberculosis (OR = 3.1, 95% CI = 1.5–6.5; *p*-value = 0.002). Microcytosis was present in 40.5% with iron deficiency, macrocytosis in 22.2% with folate deficiency, and 33.3% with vitamin B12 deficiency. The sensitivities of the reticulocyte haemoglobin content and % hypochromic red blood cells in diagnosing iron deficiency were 34.7% and 29.7%, respectively. Conclusions: HIV, iron deficiency, and tuberculosis were the most prevalent causes of moderate and severe anaemia. The majority had multiple causes. Iron, folate, and vitamin B12 deficiencies should be identified by biochemical testing rather than by red cell volume.

## 1. Introduction 

Anaemia is a significant public health problem worldwide, with adverse effects on health and socio-economic development [1]. Global anaemia prevalence in 2019 was 23.2%, affecting 1.8 billion people, resulting in an estimated 50.3 million years lived with disability [2]. Dietary iron deficiency is the most prevalent cause globally, followed by inherited hemoglobinopathies and haemolytic anaemias [2]. Iron deficiency (ID) is recognised as a significant contributor to anaemia and ill-health in patients with common chronic inflammatory conditions including chronic kidney disease (CKD) [3], heart failure [4], and cancer [5]. There is also evidence of inadequate investigation of anaemia in primary care [6,7,8]. 

Anaemia is common in South Africa, with community rates between 22% and 31% in women and 12% and 17% in men older than 15 years [9,10]. In addition, South Africa has 8.2 million persons living with human immunodeficiency virus infection (PLWHIV) [11] and an associated anaemia prevalence up to 72% [12]. South Africa also has a high tuberculosis (TB) prevalence estimated at 737 per 100,000 in 2018 [13]. Anaemia is a risk factor for TB [12,14], and TB is associated with a high prevalence of anaemia (61.5%) [15]. 

ID is probably the most common cause in South Africa [9]. Anaemia of inflammation (AI), which is synonymous with anaemia of chronic disease, is the second most common cause globally in clinical practice [16] and, most probably, in South Africa [17,18]. In persons living with HIV (PLWHIV), causes may be multifactorial, but anaemia of chronic disease is the predominant cause [19]. In Southern Africa, TB is probably the most common cause of moderate-to-severe anaemia in PLWHIV [12,20,21,22].

The relative contributions of all causes of anaemia are not well characterised in patients attending primary care and district hospitals in South Africa. In a community-based study in the Free State Province, AI was the most prevalent cause of anaemia in women with or without HIV [17]; and in a district hospital (GF Jooste) in Cape Town, >95% of patients with HIV-associated TB had AI and <3% had IDA [23]. However, no testing was done for other causes of anaemia in either study. 

The aims of this study were to determine the causes of anaemia in adults in primary care and a district hospital setting in Mthatha, South Africa. To determine the overall causes in a resource-efficient manner, the study focused on moderate and severe anaemia. 

## 2. Materials and Methods

### 2.1. Study Design

A cross-sectional study design was used. The study was undertaken between September 2017 and March 2018.

### 2.2. Study Population and Setting

The study population was patients aged ≥18 years who attended two community health centres (CHCs), one semi-urban (CHC1) and one rural (CHC2), and Mthatha Regional Hospital (MRH) general outpatients and Emergency Department managed by the Department of Family Medicine and Rural Health. MRH functions as both a district and referral hospital. The catchment area is King Sabata Dalindyebo Local Municipality, comprising a socio-economically deprived predominantly rural population [24]. 

### 2.3. Sampling

The population prevalence of moderate and severe anaemia was estimated to be ±2% [9]. This study was facility-based, and the prevalence was therefore estimated to be higher at a minimum of 3%. Using the equation n = p (100 − p) (1.96)^2^/d^2^, where n = minimum sample size, p = anticipated prevalence of moderate and severe anaemia, and d = set precision [25]; the precision was set at 2% and the estimated prevalence at 3%, yielding a minimum sample size of ±279 participants. The confidence interval was 95%. A further 20% (56) of participants was added to the sample to factor in for anticipated loss of participants during the evaluation process, loss of data, and data entry errors [26] for a total sample size of 335 participants. 

### 2.4. Study Procedures

A purposive sample was taken. Patients waiting to consult the nurse or doctor were approached by a research assistant (a nurse at each site during normal working hours). If agreeable, they signed a consent form. Exclusion criteria were: (1) pregnancy, (2) a significant bleeding episode in the last three months that needed an urgent visit to the clinic or doctor or a blood transfusion, and (3) unwillingness to disclose or test for HIV.

Entry to the study was by identifying moderate and severe anaemia using a HemoCue Hb201+ analyser^®^ on a capillary (finger-prick) sample. Anaemia was defined as mild, 11–11.9 g/dL for women and 11–12.9 g/dL for men; moderate, 8–10.9 g/dL; and severe as <8 g/dL for both sexes [27]. All patients had their HIV status assessed and a rapid HIV test done if not on antiretroviral therapy (ART). ART was endorsed for all PLWHIV from September 2016 [28]. Patients with mild anaemia were referred for routine care [29]. Those with moderate and severe anaemia had the following tests analysed at the National Health Laboratory Service (NHLS) at Nelson Mandela Central Hospital, Mthatha: full blood count, reticulocyte production index (RPI), and blood film; creatinine, eGFR (Modification of Diet in Renal Disease study equation, without the race coefficient), bilirubin (direct and indirect), C-reactive protein, vitamin B12 and folate, iron, transferrin, transferrin saturation (TSAT), and Xpert MTB/RIF test^®^ on (non-induced) sputum. In addition, patients with severe anaemia had urine and blood cultures for TB (BACTEC™ Myco/F Lytic Culture vials) and urine lateral flow lipoarabinomannan assay LF-LAM (Alere^®^). In PLWHIV, CD4 count and viral load (VL) tests were done according to national guidelines [28,30]. VL suppression was defined as < 50 RNA copies/mL [30]. Patients were also assessed by a doctor and underwent chest-X-ray and focused ultrasound examination for evidence of TB [31] and chronic kidney disease (CKD). Additional tests were ordered at the treating clinicians’ discretion. A final assessment/diagnosis was then made.

A Siemens ADVIA 2120 analyser was used initially followed by a Beckman Coulter LH 780 Analyser. The NHLS normal adult Hb values were 13–17 g/L for males and 12–15 g/L for females. Microcytosis and macrocytosis were defined as MCV <80 fL and >100 fL, respectively [32,33]. A % hypochromic red cells (%HRC) value >6% [34,35] or a reticulocyte haemoglobin content (CHr) of <29 pg/cell indicate ID [35]. 

Vitamin B12 deficiency was categorised as: <147.6 pmol/L (200 pg/mL), deficiency likely; >258.3 pmol/L (350 pg/mL), deficiency unlikely; while <73.8 pmol/L (100 pg/mL) usually indicates clinical deficiency [36]. With vitamin B12 values between 147.6–258.3 pmol/L, patient records were evaluated to determine if clinical findings were consistent with deficiency. A serum folate level <6.8 nmol/L was defined as deficiency [37].

TB was diagnosed by detection of *Mycobacterium tuberculosis* (TB) by molecular testing (Xpert MTB/RIF, Genotype^®^ MTBDRsl), urine LF LAM, culture, or demonstration of acid-fast bacilli on clinical specimens. Patients with probable TB diagnosed on X-ray or ultrasound underwent anti-tuberculosis therapy but were excluded from analysis.

IDA was defined as anaemia and a ferritin < 30 mcg/L [38,39] and in the presence of inflammation, a ferritin < 100 mcg/L, and ferritin 100–300 mcg/L if TSAT < 20% [39,40]. AI was diagnosed on three criteria: (1) a chronic inflammatory illness including cancer and haematological malignancies, infections, immune-related diseases, inflammatory diseases, CKD, heart failure, chronic pulmonary disease, and anaemia of the elderly [41]; (2) a ferritin 100–300 mcg and TSAT ≥ 20% or ferritin > 300 mcg/L [40]; and (3) the absence of an identifiable cause [16]. Guidelines for heart failure [42], cancer [43], and CKD [44] accept that inflammation is part of the disease process without a requirement for an elevated CRP. A CRP (NHLS normal value < 10 mg/L) was performed primarily to identify inflammation in people without a diagnosis of an inflammatory disorder.

CKD was diagnosed on standard criteria [45,46] and is considered the most likely cause of anaemia if the GFR < 60 mL/min/1.72 m^2^ when no other cause is identified [44,47].

### 2.5. Statistical Analysis

Data were captured in Microsoft Excel and exported to STATA version 17 for analysis. Numerical data were explored for normality using the Shapiro–Wilk test. All numerical data were not normally distributed and were summarised using the median and interquartile range (IQR = 75th percentile − (minus) 25th percentile). Categorical data were summarised using percentages. A comparison of two categorical variables was undertaken using the chi-square test if expected frequencies were ≥5, and if the expected frequencies were <5, then the Fisher’s exact test was used. A comparison of medians used the Wilcoxon rank-sum test for binary categorical variables and the Kruskal–Wallis test for nominal categorical variables (more than two groups). Predictors of anaemia severity were determined using bivariable and multivariable logistic regression analysis. The adjusted multivariable logistic regression model was determined through purposeful selection of variables to determine the best-fitting model. The odds ratio or adjusted odds ratio (OR/aOR) was the measure of association used for categorical predictors, and coefficients were used to predict numerical variables. The 95% confidence interval (95%CI) was used for the precision of estimates, and statistical significance is a *p*-value of <0.05. Because of the use of a subsample in the analyses, especially for Tables 6 and 7, the *p*-values are purely descriptive. Missing data were analysed using complete case analysis. 

## 3. Results

A total of 21 declined participation in the study. The most common reason (5) was unreadiness to know their HIV status. Table 1 summarises the demographic characteristics of patients who participated, and it shows that 1327 participants were enrolled into the study, of which 44.6% (592) were seen at MRH, and 63.5% (842) were female. The combined median age was 48 years (IQR = 31 years). There was statistical homogeneity between males and females (*p*-value = 0.100), the CD4 count (*p*-value = 0.539), and viral load (*p*-value = 0.235). The median ages of patients were statistically different depending on their recruiting health facility (*p*-value = 0.0003). Even though the combined HIV prevalence was 20.9% (277/1327), it ranged from 5.5% (22/399) in CHC1 to 35.8% (212/592) at MRH, and this was statistically significant (*p*-value < 0.0001). The HemoCue also showed anaemia prevalences of 48.4% (147/304) and 46.1% (246/534) for males and females 40 years old and older, respectively. 

Of the 1327 patients, 471 (35.5%) qualified to have a laboratory evaluation of their anaemia due to HemoCue blood levels of moderate or severe anaemia status (Table 2). However, only 463/471 (98.3%) participants had laboratory haemoglobin results, of whom 49.0% (*n* = 227) had moderate and 24.0% (*n* = 111) severe anaemia, comprising 25.5% (*n* = 338) of those screened. Chronic diseases that can cause anaemia (HIV, TB, CKD, cancer, and heart failure) and non-communicable diseases (NCDs) of high prevalence were assessed on the 471 participants who had progressed for a laboratory evaluation. The median haemoglobin levels were statistically different between the three health facilities (*p*-value = 0.002), and the difference was due to the lower median of MRH (9.4) when compared to CHC1 (10.9), which was statistically significant (*p*-value < 0.001). Disease prevalences were as follows: tuberculosis (16.6%, *n* = 78), hypertension (20.0%, *n* = 94), chronic kidney disease (5.9%, 28), and cancers (2.6%, *n* = 12). There was a statistically significant difference in the hypertension status depending on the health facility of recruitment (*p*-value < 0.001). Of those with confirmed TB, 54/78 (69.2%) had pulmonary TB, 13/78 (16.7%) extra-pulmonary TB, 7/78 (9.0%) had both, and in 4/78 (5.1%), the primary site could not be confirmed. In addition to TB, there were only two patients with CDC-Stage-3-defining illnesses [48], namely cervical cancer and cryptococcal antigenaemia. More than two-thirds of the participants with HemoCue severe and moderate anaemia who had HIV were in CDC Stage 3. The CD4 nadir, median, mean, and zenith were 1, 103, 191, and 2440, respectively. Viral replication was not suppressed in the majority of PLWHIV. 

Figure 1 illustrates the biochemical diagnosis of IDA and AI. The category AI includes those with specific causes of anaemia (e.g., vitamin deficiencies and haematological malignancies) who also had inflammation. Excluding 17 with missing ferritin results, the percentages of AI and IDA were 72.3% (232/321) and 26.5% (85/321), respectively. In patients with CKD, 8/25 (32%) had IDA. Of nine patients with cancer, 1/6 (16.7%) had IDA. Four patients with cardiac failure had AI.

Only 399 individuals had a full record of the time from venesection to FBC analysis. The time a specimen was received at the laboratory was used as a proxy for the time to analysis, as reports did not specify analysis times. With a median of 2.6 h (IQR = 2.2 h), all FBC analyses were undertaken within 24 h. Figure 2 shows no statistically significant difference in the median MCV in those analysed <8 h and ≥8 h (*p* = 0.766) and in the median %HPO for those analysed <6 h and ≥6 h (*p* = 0.176). 

Using RHC to diagnose IDA, as shown in Table 3, sensitivity was 34.7%, specificity 89.4%, and positive predictive value 64.7%.

Table 4 shows the use of HRC to diagnose IDA. Sensitivity was 29.7%, specificity 87.9%, and positive predictive value 92.2%.

Table 5 shows that patients with ID had predominantly normocytic anaemia, as also did patients with folate deficiency. Patients with vitamin B12 deficiency had equal proportions of microcytic, normocytic, and macrocytic anaemia. In patients 65 years old and older with IDA, 35.3% (6/17) had microcytosis.

In eight patients with vitamin B12 deficiency, two had neurological disorders consistent with deficiency. One had acute confusion (vitamin B12, 50 pmol/L) and one lumbar radiculopathy (vitamin B12, 75 pmol/L). They had normal MCV and no blood film features of megaloblastic anaemia. Another patient was on metformin, which may cause deficiency. Five had no neurological or haematological features to diagnose deficiency. Of three patients with possible vitamin B12 deficiency (range 147.6–258.3 pmol/L), one had acute psychosis with HIV and hepatitis B virus infections, one had paraparesis of unknown cause, and one PLWHIV had TB meningitis. Overall, 2.5% (8/324 results) had vitamin B12 deficiency. 

In patients with moderate and severe anaemias and a vitamin B12 level <147.6 pmol/L, none had hyper-segmented neutrophils or Howell–Jolly bodies. One patient had macrocytosis and ovalocytes with a vitamin B12 level of 109 pmol/L and a folate level of 2.1 pmol/L. The prevalence of folate deficiency was 6.1% (18/294 results). None of the patients with folate deficiency were taking a medication causing deficiency.

### 3.1. Macrocytosis

Of 25 patients with macrocytosis, 10 had incomplete drug data and 15 were PLWHIV. The putative causes of macrocytosis in 15 were lamivudine, 8; folate deficiency, 3; vitamin B12 deficiency, 2; cotrimoxazole, 2; valproate, 2; zidovudine, 2; liver disease, 1; and multiple myeloma, 1 (four patients had two causes each, and one had three). 

### 3.2. Microcytosis

Of 78 patients with microcytosis, one had a Mentzer index value < 13. That patient had IDA (TSAT 4%, ferritin 46 mcg/L). 

### 3.3. Multiple Causes of Anaemia

For each of the eight major causes, the numbers (percentage) of patients with moderate/severe anaemia due solely to that cause were iron deficiency 32/85 (37.6), vitamin B12 deficiency 4/8 (50), folate 1/18 (5.6), HIV 136/219 (62.1), cancer 2/12 (16.7), CKD 8/28 (28.6), TB 6/75 (8.0), and heart failure 2/4 (50). Overall, 191/449 (42.5%) had a single cause of anaemia, ranging from 5.6% for patients with folate deficiency to 62.1% with HIV. The remainder (57.5%) had two or more causes. 

In the bivariable model, those who had severe anaemia were 70% (OR = 1.7, 95%CI = 1.0–2.7; *p*-value = 0.039) and 90% (OR = 1.9, 95%CI = 1.1–3.2; *p*-value = 0.017) more likely to have HIV and TB, respectively (Table 6). However, the multivariable best-fitting model that was adjusted for other chronic conditions and the anaemia markers did not show an association between the severity of anaemia and an HIV-positive result (*p*-value = 0.880). Instead, the measure of effect increased for TB, wherein patients with severe anaemia were found to have been three times more likely to have had TB, and this was also statistically significant (OR = 3.1, 95% CI = 1.5–6.5; *p*-value = 0.002). Similarly, patients with severe anaemia were 5.5 times more likely to present in a hospital setting than a primary care setting, and this was statistically significant as well (OR = 5.5, 95%CI = 1.5–19.7; *p*-value = 0.009).

The multivariable analysis further shows that an increase of 1 in the reticulocyte production index increased the odds of severe anaemia by 5.5 (co-efficient = 1.7, 95%CI = 0.9–2.5; *p*-value < 0.0001); and a 1 g/dL increase in the mean corpuscular haemoglobin concentration (MCHC) increases the odds of having severe anaemia by 65% (co-efficient = 0.5, 95%CI = 0.4–0.7; *p*-value < 0.0001), and both these were statistically significant.

Even though PLWHIV with moderate anaemia had a statistically lower median Hb than HIV-negative patients (*p*-value < 0.0004), there was no statistical difference between the median Hb of patients with a negative HIV status and PLWHIV in patients with severe anaemia (*p*-value = 0.623) (Table 7). TB showed an opposite effect where patients with severe anaemia had a higher median Hb, and this was statistically significant (*p*-value = 0.005). The median MCHC was lower than normal in patients with moderate and severe anaemia and lower still in patients with severe anaemia and HIV or TB. Other markers of statistical significance included the following: a higher median CRP for PLWHIV with moderate anaemia (*p*-value = 0.003); higher median CRP values for both TB-positive patients with moderate (median = 146; *p*-value = 0.005) and severe anaemia (median = 163; *p*-value = 0.0004); higher median transferrin saturation severe anaemia in both HIV-positive (median = 19.6%; *p*-value = 0.0004) and TB-positive patients (median = 19.2%; *p*-value = 0.015); lower median RPIs for PLWHIV with moderate (median = 0.5; *p*-value = 0.005) and severe anaemia (median = 0.3; *p*-value = 0.036); and higher median ferritin levels for all categories of HIV-positive and TB-positive patients (*p*-value < 0.05). Vitamin B12 values were higher in PLWHIV with moderate or severe anaemia and in TB with moderate anaemia. Folate levels were unchanged with HIV and TB. 

## 4. Discussion

There was an overall high (45.9%) prevalence of anaemia in the screened population, with the highest prevalence at MRH. Similar to attendees in primary care in South Africa [49], two-thirds of patients screened for anaemia in this study were female. In this study, the age group 40–49 years had the largest number screened. This may be due to the exclusion of pregnant women and the inclusion of sicker patients attending MRH. In older adults (≥40 years old), 48.4% of men and 46.1% of women had anaemia. High rates of anaemia were also found in community-living older adults in Mpumalanga Province, South Africa, i.e., 40.1% of men and 43% of women [50]. Prevalence of HIV infection was similar, at 20.9% in this study and 20.7% in Mpumalanga Province.

Overall, the largest category of anaemia on screening was moderate, followed by severe. In contrast, in the national community surveys, the largest category was mild followed by moderate [9,10]. The skewed distribution in this study is most likely due to selection of participants from health care facilities, which will likely have more sicker individuals than those in the general population. The overall prevalence of moderate and severe anaemia was 35.5% on HemoCue testing but 25.5% on laboratory testing. HemoCue 201 overestimates mean Hb concentrations by 0.1–1.2 g/dL [51]. Furthermore, capillary finger-prick usually produces higher Hb values by 0.2–0.9 g/dL compared to venous blood [51]. 

In patients with laboratory-confirmed moderate and severe anaemia, AI was the predominant type at 72.6%, reflecting the high prevalences of HIV (59.5%) and TB (16.6%). In Cape Town, >95% of patients with anaemia and HIV-associated TB had AI [23]. In patient populations where there are very low levels of HIV and TB diagnoses, AI was the commonest type of anaemia, but prevalences were much lower at 25.7% in elderly inpatients in South Africa [18] and 41.9% of medical inpatients in Italy [52]. In the U.S., one-third of elderly community-dwellers with anaemia had AI, one-third nutrient deficiencies, and one-third unexplained anaemia [53].

The 26.5% prevalence of IDA is similar to prevalences of 24.3% in older inpatients in South Africa [18] and 20% in older community-dwellers in the U.S. [53] but higher than 14.7% in internal medicine inpatients in Italy [52]. IDA in this study comprised 15.9%, with absolute ID (ferritin < 100 mcg/L [54]) and 10.5% functional ID (ferritin ≥ 100–300 mcg/L and TSAT < 20%) [39,43]. However, absolute and functional ID overlap such that patients who have ferritin > 100 mcg/L may have absolute iron deficiency. In a systematic review utilising 38 studies, the mean ferritin level in absolute iron deficiency was 82.4 mcg/L (range of means 34–158 mcg/L) in patients with inflammatory diseases, using bone marrow iron as the gold standard [55]. While there are various definitions of ID [23], the “pragmatic” definition of Camaschella and Girelli [39] was used in this study, combining ferritin and TSAT. TSAT is considered an accurate test for ID in patients with inflammation [39,56]. Newer tests for ID include hepcidin and soluble transferrin receptor (sTfR) levels [40] but are not routinely available in South Africa.

Forty-one per cent (41%) of patients with IDA had microcytosis. In patients 65 years old and older, 35.3% had microcytosis. Studies in older patients with absolute IDA in developed countries show <30% have microcytosis [57]. These data strongly support biochemical testing for ID in all patients with anaemia irrespective of MCV values. Only one with microcytosis had a Mentzer index value < 13 [58] but had concomitant ID. This suggests that the thalassaemia trait is uncommon in the study population. While the α-thalassaemia trait is present in some communities in South Africa, with a prevalence of 3.8% [59] and 16% in non-random samples [60], testing is mainly done if there is unexplained microcytosis [60,61]. 

All MCV measurements in the laboratory were done within 24 h of venesection, with no difference between those measured before and after eight hours. While MCV stored at 4 °C remains unchanged for 24 h, it increases significantly after eight hours at room temperature (standard mean difference −0.30, CI −0.50, −0.10)) [62]. 

RHC and %HYPO had low sensitivity for IDA, implying that the tests are not suitable for ruling out IDA in the study setting. RHC can perform as well as standard tests for the diagnosis of ID [35]. %HPO has mainly been used to diagnose ID in the setting of CKD [35,39]. While %HPO should be analysed within six hours of venesection [34], Figure 2 shows that there was no significant difference between blood samples drawn before or after six hours. Further research is needed into the sub-optimal performance of these tests in similar settings to this study.

The major disease categories associated with moderate and severe anaemia were, in order, HIV, TB, CKD, and cancer, as shown in Table 2. Anaemia is common in PLWHIV from multifactorial causes [63]. While worldwide prevalences in PLWHIV are 21.6%, 22.6%, and 6.2% for mild, moderate, and severe anaemia, respectively [64], in South Africa, prevalences are mostly higher at 26.7%, 41.1%, and 4.3% [12]. In this study, 59.5% of those with moderate/severe anaemia were PLWHIV. The majority had low mean CD4 counts, unsuppressed VLs, and CDC Stage 3, which are indicative of advanced HIV disease and a high risk of anaemia. The low prevalence of reported OIs other than TB in this study may be due to limited investigations. Compared to low- and middle-income countries in Latin America and Asia, adults in Sub-Saharan Africa have markedly lower incidences of *Pneumocystis jirovecii* pneumonia and cerebral toxoplasmosis in ART-naïve patients [65], most likely due to a lack of resources for diagnosis rather than a true difference in infection rates [66].

Patients with severe anaemia were three times more likely to have a diagnosis of TB compared to the other major diagnostic categories (HIV, cancer, and CKD) as demonstrated in the multivariate analysis (Table 6, which only considers patients with laboratory moderate or severe anaemia). While Table 7 shows that in patients with TB and severe anaemia, Hb was higher in those with TB compared to those without, the multivariate analysis is the most accurate summation due to accounting for multiple variables. Table 7 is an analysis of indices to assist in characterising findings in HIV and TB, which are the most prevalent causes, compared to all other causes in patients with moderate/severe anaemia. In PLWHIV with moderate (but not severe) anaemia, Hb was significantly lower compared to those without HIV. HIV frequently causes impaired haematopoiesis, and anaemia is the most common manifestation, increasing in frequency and severity with disease progression [63]. PLWHIV with moderate or severe anaemia had overall low CD4 counts (median values < 200 cells/µL) indicative of more advanced disease [67]. PLWHIV with TB and severe anaemia had lower CD4 counts and higher VL compared to those without TB. This is consistent with HIV as the most potent immunosuppressive risk factor for active TB [68]. HIV infection was associated with a slightly lower RPI that is not clinically significant. Patients with or without HIV or TB and moderate and severe anaemia had normal MCV and low MCHC (indicating hypochromia). However, MCHC is considered of little clinical relevance in interpreting anaemias [69]. Median TSAT was also very low (<16%) in both moderate and severe anaemia without HIV or TB, which is suggestive of absolute ID [56]. Serum iron was lower in severe anaemia without HIV or TB. Serum iron is depressed in both ID and AI and cannot differentiate between the two conditions [70]. Serum transferrin was significantly lower in patients with moderate and severe anaemia and HIV or TB. It is an acute-phase reactant that deceases in inflammation but increases in ID [41]. Ferritin and CRP, both inflammatory markers, were significantly higher in patients with either HIV or TB and moderate or severe anaemia. CRP can be used for diagnosis of pulmonary TB [71] and as an indicator of disease severity [72] and higher mortality [73]. Median CRP levels in patients with TB in this study, i.e., 146 mg/L (IQR 214 mg/L) in moderate and 163 mg/L (IQR 122 mg/L) in severe anaemia, were considerably higher than those of 40 mg/L (IQR 83 mg/L) in U.K. [72] and 17.3 (±37.2 mg/L, SD) in Taiwanese [73] studies and indicative of more severe disease, possibly due to delayed diagnoses.

CKD stages 3–5 were present in 6.5% of patients with moderate and severe anaemia. Prevalences of CKD stages 3–5 in Africa are 4.6% in the general population and 13% in high-risk populations, e.g., HIV, diabetes, and hypertension [74], compared with 10.6% in the general population worldwide [75]. The low prevalence of CKD may be due to the strict application of the criteria to define CKD, i.e., abnormalities of kidney structure or function, present for >3 months [46], and non-performance of regular renal evaluations, in patients at high risk, e.g., diabetes mellitus as elsewhere in South Africa [76], and the non-availability of prior patient records with creatinine results. The actual prevalence is most likely double this, as all creatinine-based GFR-estimating equations underestimate GFR in African populations [77]. While cystatin C-based equations may be superior, cystatin C is unaffordable in clinical practice in developing countries [78]. In this study, 32% with CKD had IDA. This compares to a 35.3% IDA prevalence in non-dialysis CKD outpatients in Johannesburg [79] and 15–72.8% in developed countries [80]. It is important that IDA is recognised and treated to improve quality of life [81]. Most patients with stable CKD can be managed in primary care [82]. 

In patients with cancer, one of six had IDA. The prevalences of anaemia and ID in different cancers was reported as 29–46% and 7–42%, respectively, in developed countries (ID prevalence at 60% was highest in colorectal cancer) [5]. No patient had colorectal cancer in this study.

There was a low overall prevalence (2.5%) of vitamin B12 deficiency (<147.6 pmol/L) compared to 8.8% in non-pregnant women of childbearing age elsewhere in South Africa [83]. While NHLS normal values for both sexes were 133–675 pmol/L, the lower limit of normal cannot be used as a cut-off to define deficiency due to frequent false-positive and false-negative results [36] Methylmalonic acid and homocysteine levels are required to confirm vitamin B12 deficiency [36], but testing was not mandated in the public-sector guidelines [84]. As seen in this study, neuropsychiatric disorders due to deficiency often occur without haematological changes [36]. While vitamin B12 deficiency can be found in up to 30% of PLWHIV [85], there is little evidence of overt clinical disease [86,87]. A minority of patients with vitamin B12 deficiency exhibited macrocytic changes. This is well documented, particularly in patients with iron deficiency or inflammation [88]. Vitamin B12 values were significantly higher in patients at MRH compared to the CHCs. Levels were higher also in moderate and severe anaemia in PLWHIV and patients with TB. There are reports of higher mortality with elevated vitamin B12 levels, namely >250 pmol/L [89] and >700 pmol/L [90] by mechanisms yet to be elucidated.

Low community folate-deficiency prevalences of 0.2% [91] and zero in reproductive-age women [83] were reported in South Africa following folate fortification in 2003. The higher prevalence (6.2%) in this study may be due to inadequate nutrition during illness.

The majority (57.5%) of patients had multiple causes of anaemia compared with 13.6% of elderly inpatients in South Africa [18] and 32.5% in Italy [52]. This relates to the high levels of infectious diseases (TB and HIV) in this study population. 

Approaches in the evaluation of anaemia are traditional evaluation using the MCV and RPI [92,93] and biochemical tests [16,23,40]. South African national guidelines advise categorising anaemia based on MCV and, if microcytic, testing for ID and, if macrocytic, testing for vitamin b12 and folate deficiency [33,94]. As shown in this and other studies [95], this approach will miss many patients with nutrient deficiencies and those with multiple causes. With a substantial prevalence of ID, this study supports the recommendation that iron status needs evaluation in every patient with anaemia [16,32] but also adds the evaluation of folate and vitamin B12 deficiencies. In areas of high TB and HIV prevalence, TB (pulmonary and extra-pulmonary) should be excluded as a cause of moderate and severe anaemia [12,22]. Overall, a combination of MCV/RPI, biochemical, HIV, and TB testing would seem the optimal approach for moderate and severe anaemia evaluation in the study population. 

The strengths of this study are that causes of anaemia were investigated comprehensively, using biochemical criteria and clinical assessment, and in the context of daily practice in primary care and a district hospital setting. An updated pragmatic definition of ID was used compared with previous studies. The cross-sectional study design is suited to estimate the prevalence of anaemia in the study population [96].

Limitations include missing data due to patient, research assistant, and laboratory factors. Patients had multiple non-interoperable records, making it difficult to obtain historical data. However, there was no indication of a systematic pattern that might invalidate the analysis. Some patients were on drugs associated with anaemia. Due to the cross-sectional study design (with absence of follow-up), it is not possible to state with certainty that any drug was responsible for anaemia. TB was likely responsible for more causes of anaemia, as patients with probable TB on clinical assessment were excluded from analysis. In retrospect, our sample size could have been optimised by using the two-proportions formula with a power of 0.8 instead of using the single-proportion formula, which resulted in only 25.5% of the initial participants being eligible for the final analyses.

The findings are not generalisable but may be transferrable to similar settings.

## 5. Conclusions

In summary, HIV, ID, and TB were the most prevalent causes of moderate and severe anaemia. Anaemia of inflammation was the most common type of anaemia. Patients with severe anaemia were three times more likely to have TB. Changes in RBC indices do not reliably predict nutritional deficiencies. Biochemical tests should routinely be performed to assess iron, folate, and vitamin B12 deficiencies.

## Figures and Tables

**Figure 1 ijerph-20-03584-f001:**
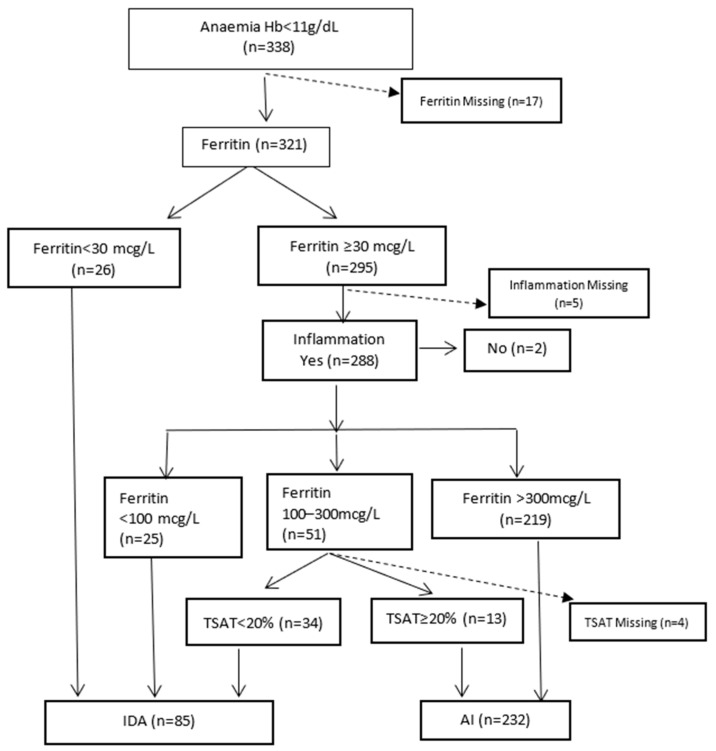
Diagnosis of iron-deficiency anaemia and anaemia of inflammation. Cappellini et al. [40] and Camaschella and Girelli, 2020 [39].

**Figure 2 ijerph-20-03584-f002:**
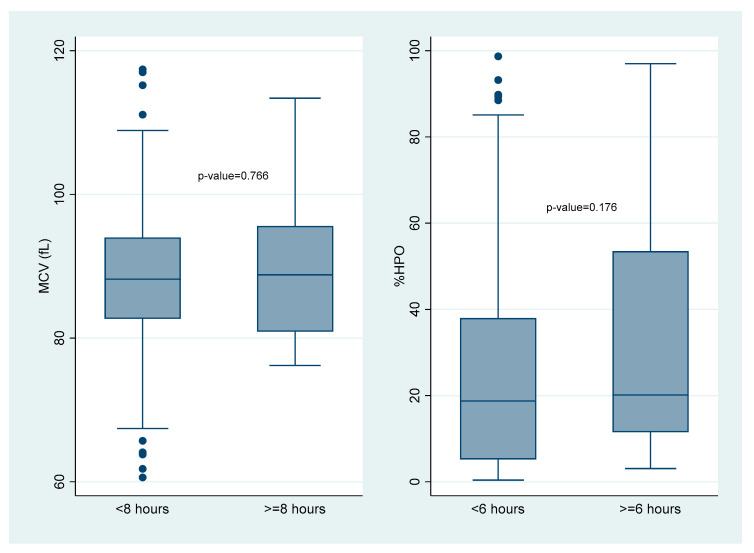
Duration of time to analysis of blood specimen for MCV and HPO.

**Table 1 ijerph-20-03584-t001:** Participant demographic characteristics and HemoCue Hb.

Characteristics	MRH		CHC1		CHC2		Total		*p*-value
	*n* = 592		*n* = 399		*n* = 336		*n* = 1327		
Sex; *n* (%)									
Female	362	(61.2)	251	(62.9)	229	(68.2)	842	(63.5)	0.100 *
Male	230	(38.9)	148	(37.1)	107	(31.9)	485	(36.6)	
Age, years; med (p25–p75)	45	(31.5–60.0)	53	(37.0–67.0)	48.5	(34.0–64.0)	48	(33.0–64.0)	0.0003 ^ʊ^*
Age categories, years; *n* (%)									
18–29	101	(18.8)	58	(14.5)	50	(14.9)	219	(16.5)	0.011 *
30–39	131	(22.1)	55	(13.8)	70	(20.8)	256	(19.3)	
40–59	188	(31.8)	147	(36.8)	107	(31.9)	442	(33.3)	
60–74	103	(17.4)	82	(20.6)	68	(20.2)	253	(19.1)	
≥75	59	(10.0)	57	(14.3)	41	(12.2)	157	(11.8)	
HIV; *n* (%)									
Positive	212	(35.8)	22	(5.5)	43	(12.8)	277	(20.9)	<0.0001 *
Negative	380	(64.2)	377	(94.5)	293	(87.2)	1050	(79.1)	
CD4, cells/µL; med (p25–p75)	101.5	(38–246.0)	48.5	(15.5–413.0)	150	(45–246.5)	101.5	(36–252.0)	0.539 ^ʊ^*
VL, log(copies/mL); med (p25–p75)	3.6	(1.8–5.7)	4.2	(2.4–5.9)	4.8	(3.0–5.7)	3.8	(1.9–5.7)	0.235 ^ʊ^*
Anaemia, HemoCue; *n* (%)									
Normal	168	(28.3)	292	(73.2)	224	(66.7)	684	(51.5)	<0.0001 *
Mild	86	(14.5)	45	(11.3)	41	(12.2)	172	(13.0)	
Moderate	190	(32.0)	50	(12.5)	50	(14.9)	290	(21.9)	
Severe	148	(25.0)	12	(3.0)	21	(6.3)	181	(13.6)	
Hb-HemoCue, g/dL; med (p25–p75)	10	(8.0–12.5)	13	(12.0–14.1)	12.5	(11.2–13.5)	12	(9.6–13.5)	0.0001 ^ʊ^*

p75, 75th percentile minus; p25, 25th percentile; med, median; VL, viral load; Hb, haemoglobin; ^ʊ^* * Chi-squared test was used; Kruskal–Wallis test was used.

**Table 2 ijerph-20-03584-t002:** Clinical conditions of participants with haemoglobin <11 g/dL by HemoCue.

Characteristics	MRH		CHC1		CHC2		Total		*p*-Value
	*n* = 338		*n* = 62		*n* = 71		*n* = 471		
Laboratory Hb **, g/dL; med (p25–p75)	9.4	(7.8–10.9)	10.9	(9.5–11.9)	9.7	(8.1–10.7)	9.7	(8.0–11.1)	<0.001 ^ʊ^*
Laboratory Hb **; *n* (%)									
Normal	32	(9.5)	13	(21.0)	7	(9.9)	52	(11.0)	0.002 *
Mild	49	(14.5)	17	(27.4)	7	(9.9)	73	(15.5)	
Moderate	163	(48.2)	25	(40.3)	39	(54.9)	227	(48.2)	
Severe	89	(26.3)	6	(9.7)	16	(22.5)	111	(23.6)	
Missing	5	(1.5)	1	(1.6)	2	(2.8)	8	(1.7)	
Vitamin B12 ^ȹ^, pmol/L; med (p25–p75)	562	(360.0–1051.0)	383	(280.0–709.0)	425.5	(292.0–568.0)	492	(333.0–943.0)	<0.001 ^ʊ^*
Folate ^ж^, nmol/L; med (p25–p75)	18.4	(11.3–27.5)	18.7	(11.3–31.9)	23.5	(15.8–32.7)	19.1	(12.0–29.7)	0.019 ^ʊ^*
Transferrin saturation ^ß^*, %; med (p25–p75)	16.0	(10.0–28.0)	17	(12.0–21.0)	13	(7.0–19.0)	16	(10.0–25.0)	0.024 ^ʊ^*
Ferritin ^ȹ^, µg/L; med (p25–p75)	678.5	(222.0–1569.0)	177.5	(69.0–520.0)	154	(49.0–1686.0)	520	(139.0–14206.0)	<0.001 ^ʊ^*
HIV; *n* (%)									
Positive	201	(59.5)	18	(29.0)	41	(57.8)	260	(55.2)	
Negative	137	(40.5)	44	(71.0)	30	(42.3)	211	(44.8)	<0.001 *
CD4 ^ȸ^*, cells/µL; med (p25–p75)	100	(37–252)	67.5	(21–413)	150	(43–242)	103	(36–255)	0.866
CD4 ^ȸ^*, cells/µL; *n* (%)									
<200 (Stage 3)	125	(67.6)	10	(62.5)	21	(61.8)	156	(66.4)	0.492 ***
200–499 (Stage 2)	42	(22.7)	3	(18.8)	11	(32.4)	56	(23.8)	
≥500 (Stage 1)	18	(9.7)	3	(18.8)	2	(5.9)	23	(9.8)	
Viral load; *n* (%)									
<50 copies/mL	29	(19.5)	1	(7.7)	3	(10.7)	33	(17.4)	0.455 ***
≥50 copies/mL	120	(80.5)	12	(92.3)	25	(89.3)	157	(82.6)	
RPI, count; med (p25–p75)	0.5	(0.3–0.9)	0.7	(0.5–1)	0.6	(0.3–0.8)	0.6	(0.3–0.9)	0.003 ^ʊ^*
MCHC, g/dL; med (p25–p75)	32.2	(30.7–33.3)	31.8	(30.9–32.9)	31.8	(30.3–32.7)	32	(30.7–33.1)	0.016 ^ʊ^*
Diabetes mellitus ^ȸ^; *n* (%)									
Yes	29	(8.7)	6	(9.7)	10	(14.1)	45	(9.6)	0.370 *
No	306	(91.3)	56	(90.3)	61	(85.9)	423	(90.4)	
Hypertension ^#^; *n* (%)									
Yes	49	(14.5)	29	(46.8)	16	(22.5)	94	(20.0)	<0.001 *
No	288	(85.5)	33	(53.2)	55	(77.5)	376	(80.0)	
Tuberculosis; *n* (%)									
Yes	58	(17.2)	7	(11.3)	13	(18.3)	78	(16.6)	0.051 ***
No	193	(57.1)	36	(58.1)	48	(67.6)	277	(58.8)	
Contaminated	3	(0.9)	2	(3.2)	2	(2.8)	7	(1.5)	
Missing ^¶^	84	(24.9)	17	(27.4)	8	(11.3)	109	(23.1)	
Chronic kidney disease; *n* (%)									
Yes	22	(6.5)	3	(4.8)	3	(4.2)	28	(5.9)	0.703 ***
No	316	(93.5)	59	(95.2)	68	(95.8)	443	(94.1)	
Cancer ^#^; *n* (%)									
Yes ^^^	9	(2.7)	1	(1.6)	2	(2.8)	12	(2.6)	1.000 ***
No	325	(96.4)	61	(98.4)	69	(97.2)	455	(96.8)	
Probable ^Ҧ^	3	(0.9)	0	(0)	0	(0)	3	(0.6)	
Heart failure ^#^; *n* (%)									
Yes	4	(1.2)	2	(1.7)	0	(0)	6	(1.3)	
No	333	(98.8)	60	(96.8)	71	(100.0)	464	(98.7)	0.293 ***

^#^ missing = 1; ** missing = 8; ^ȸ^ missing = 3; ^ȸ^* missing = 25; ^ж^ missing = 69; ^ȹ^ missing = 35; ^ß^* missing = 78; ^^^ Prostate cancer = 3; oesophageal cancer and breast cancer = 2; cervix, squamous carcinoma neck, adenocarcinoma lung, multiple myeloma, and acute myeloid leukaemia= 1; ^Ҧ^ probable bone cancer, undetermined pelvic mass, and probable lymphoma = 1; ^¶^ excludes those with probable TB based on X-ray diagnosis. p75, 75th percentile; p25, 25th percentile; med, median; Hb, haemoglobin. * Chi-squared test was used; *** Fisher’s exact test was used; ^ʊ^* Kruskal–Wallis test was used.

**Table 3 ijerph-20-03584-t003:** Iron-Deficiency Anaemia and Reticulocyte Haemoglobin Content.

Iron-Deficiency Anaemia	RHC < 29		RHC > −29		Total	
	*n*	%	*n*	%	N	%
Yes	33	(34.7)	18	(10.6)	51	(19.2)
No	62	(65.3)	152	(89.4)	214	(80.9)
Total	95	(100.0)	170	(100)	265	(100)

**Table 4 ijerph-20-03584-t004:** Iron-Deficiency Anaemia and % Hypochromic Red Cells.

Iron-Deficiency Anaemia	%HPO Analysis				Total	
	≥6%		<6%			
Yes *n* (%)	47	(92.2)	4	(7.8)	51	(100.0)
No; *n* (%)	111	(79.3)	29	(20.7)	140	(100.0)
Total; *n* (%)	158	(82.7)	33	(17.3)	191	(100.0)

**Table 5 ijerph-20-03584-t005:** Iron, Folate, and Vitamin B12 deficiency and Mean Corpuscular Volume in laboratory moderate/severe anaemia.

Deficiency	Microcytic	Normocytic	Macrocytic	TOTALS
Iron; * *n* (%)	34 (40.5)	47 (56.0)	3 (3.6)	84 (100.0)
Folate; *n* (%)	3 (16.7)	11 (61.1)	4 (22.2)	18 (100)
Vitamin B12; *n* (%)	3 (33)	3 (33)	2 (33)	8 (100.0)

Iron-deficiency anaemia: ferritin <30 mcg/L and with inflammation, <100 mcg/L, or 100–300 mcg/L plus TSAT < 20% [39]. A folate level <6.8 nmol/L defined deficiency [37]. Vitamin B12 deficiency was defined as <147.6 pmol/L (200 pg/mL) [36]. * One patient with IDA had a missing MCV result.

**Table 6 ijerph-20-03584-t006:** Bivariable and Multivariable predictors of severe anaemia among patients with moderate or severe anaemia.

Variable		Bivariable Analysis			Multivariable Analysis		
	*n* (N = 338 ^^^*)	OR/ß	95% CI	*p*-Value	aOR/ß	95% CI	*p*-Value
HIV							
Negative	33/119	ref	-	1	ref	-	1
Positive	78/219	1.7	(1.0–2.7)	0.036	0.9	(0.4–2.0)	0.880
TB							
No	66/219	ref	-	1	ref	-	1
Yes	34/75	1.9	(1.1–3.2)	0.017	3.1	(1.5–6.5)	0.002
Cancer							
No	107/322	ref	-	1	ref	-	1
Yes	4/12	1.1	(0.3–3.8)	0.871	3.9	(0.6–24.7)	0.153
Chronic kidney disease							
No	102/310	ref	-	1	ref	-	1
Yes	9/28	1.1	(0.5–2.4)	0.878	0.8	(0.3–2.7)	0.763
Health facility							
CHC1	6/31	ref	-	1	ref	-	1
CHC2	16/55	2.1	(0.7–5.9)	0.323	2.0	(0.5–8.2)	0.324
MRH	89/252	2.9	(1.2–7.1)	0.022	5.5	(1.5–19.7)	0.009
Sex							
Female	73/229	ref	-	1	ref	-	1
Male	38/109	1.2	(0.7–1.9)	0.481	1.1	(0.5–2.2)	0.789
MCHC ^ß^	7/168	0.3	(0.2–0.4)	<0.0001	0.5	(0.4–0.7)	<0.0001
RPI ^ß^	6/154	1.3	(0.6–1.9)	<0.0001	1.7	(0.9–2.5)	<0.0001
Vitamin B12 * 1000 ^ß^	7/164	−0.4	(−0.8; −0.04)	0.030	−0.7	(−1.2; −0.07)	0.028
Transferrin saturation * 20 ^ß^	7/147	0.001	(−0.02; 0.004)	0.645	-	-	-
C-reactive protein * 100 ^ß^	6/161	0.01	(−0.2; 0.2)	0.901	-	-	-
MCV ^ß^	7/169	0.04	(0.02–0.1)	0.001	-	-	-
Ferritin * 10,000 ^ß^	7/164	−0.01	(−0.5; 0.5)	0.966	-	-	-
Platelets * 100 ^ß^	7/171	−0.05	(−0.2; 0.1)	0.457	-	-	-

OR, odds ratio; aOR, adjusted odds ratio; ^^^* complete cases in multivariable model = 266; ^ß^ coefficients used; ref, reference category; 95%CI, 95% confidence interval; MRH, Mthatha Regional Hospital; *n*, number of patients with severe anaemia; N, Total number of patients with severe and moderate anaemia.

**Table 7 ijerph-20-03584-t007:** Median and interquartile range (IQR) comparison by HIV and TB status for participants with laboratory severe and moderate anaemia.

Variable	HIV						Tuberculosis					
	Moderate			Severe			Moderate			Severe		
	HIV Positive	HIV Negative	*p*-Value	HIV Positive	HIV Negative	*p*-Value	TB Positive	TB Negative	*p*-Value	TB Positive	TB Negative	*p*-Value
Hb, g/dL; med (IQR)	9.4 (1.4)	9.9 (1.3)	0.0004	6.7 (1.9)	6.4 (2.0)	0.623	9.7 (1.5)	9.5 (1.4)	0.939	7.1 (1.7)	6.1 (2.0)	0.005
MCV, fL; med (IQR)	88.9 (11.2)	87.9 (11.3)	0.610	86.2 (14.8)	83.5 (18.6)	0.279	86.1 (7.8)	88.6 (11.8)	0.156	85.7 (14.7)	85.4 (18.3)	0.963
MCHC, g/dL; med (IQR)	31.7 (2.3)	31.2 (1.9)	0.080	30.5 (2.2)	29.2 (4.7)	0.016	31.9 (2.1)	31.5 (2.2)	0.561	31.1 (2.6)	29.9 (3.4)	0.003
HPO, %; med (IQR)	15.8 (21.0)	20.6 (31.2)	0.063	28.9 (41.3)	68.4 (33.3)	0.010	10.6 (16.3)	20.2 (29.8)	0.142	31.9 (40.2)	59.1 (52.0)	0.031
RHC, pg; med (IQR)	31.3 (6.9)	30.8 (4.0)	0.524	31.0 (6.3)	25.2 (7.3)	0.004	30.2 (5.3)	31.0 (5.6)	0.673	29.9 (4.9)	29.4 (9.8)	0.297
RPI, count; med (IQR)	0.5 (0.6)	0.7 (0.5)	0.005	0.3 (0.3)	0.4 (0.5)	0.036	0.6 (0.6)	0.5 (0.5)	0.127	0.3 (0.3)	0.4 (0.3)	0.146
Platelets, * 10^9^/L; med (IQR)	307.0 (182)	331 (211)	0.110	294.0 (200)	355.0 (225.0)	0.051	357.0 (227)	314.5 (189.5)	0.483	252.0 (202.0)	330.0 (201.0)	0.059
Iron, µg/dL; med (IQR)	305.1 (299.3)	351.1 (236.0)	0.186	284.9 (316.6)	189.9 (184.2)	0.008	293.5 (184.2)	345.3 (316.6)	0.133	290.7 (293.5)	207.2 (241.7)	0.032
Transferrin, mg/dL; med (IQR)	1380 (790)	1920 (1080)	0.0001	1305 (850)	1560 (1150)	0.010	1350 (570)	1685 (1070)	<0.012	1160 (580)	1475 (1320)	0.009
Transferrin saturation (%); med (IQR)	16.5 (18.2)	14.5 (11.3)	0.171	19.6 (21.0)	7.3 (15.5)	0.0004	16.5 (9.3)	14.8 (15.3)	0.858	19.2 (20.0)	12.2 (19.6)	0.015
Ferritin, µg/L; med (IQR)	712.0 (1590.0)	423 (880.5)	0.003	1376.0 (1771.0)	162.0 (912.0)	0.0001	925 (1390.0)	458.0 (1153.0)	0.0004	2000.0 (1751.0)	550.0 (1349.0)	0.0001
CRP, mg/L; med (IQR)	103.5 (198.0)	41.5 (129.0)	0.003	109.5 (148.0)	92.0 (143.0)	0.462	146.0 (214)	59.0 (162.0)	0.005	163.0 (122.0)	86.5 (132.5)	0.0004
Vitamin B12, pmol/L; med (IQR)	522.5 (735.5)	416.0 (528)	0.009	767.5 (957.0)	530.0 (450.0)	0.043	688.0 (660.0)	445.0 (587.0)	0.002	785.0 (1193.0)	595.0 (544.0)	0.129
Folate; nmol/L; med (IQR)	18.6 (17.8)	21.5 (14.8)	0.081	16.7 (16.0)	18.9 (23.9)	0.103	18.2 (16.9)	21.1 (16.9)	0.150	17.5 (15.3)	18.9 (23.6)	0.631
CD4, cells/µL; med (IQR)	103.5 (213.0)	-	-	72.0 (189.0)	-	-	68 (200.0)	114.0 (213.0)	0.178	47.0 (98)	124.0 (193.0)	0.013
VL, log(copies/mL); med (IQR)	4.0 (3.7)	-	-	3.8 (3.3)	-	-	4.3 (3.2)	4.0 (3.8)	0.242	5.3 (3.2)	2.9 (2.7)	0.001

IQR, 75th percentile minus (−) 25th percentile; dL, decilitre; RHC, reticulocyte haemoglobin content; RPI, reticulocyte production index. The National Health Laboratory Service in South Africa reports iron in µmol/L and transferrin in g/L. Normal ranges: MCHC, males 33–35, females 32.7–34.9 g/dL; Iron-Deficiency Indices: %HYPO > 6%; RHC < 29 pg/cell. The Wilcoxon rank-sum test was used.

## Data Availability

The data presented in this study are available in article or Appendix A.

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
