# Peer review of "Causes of Moderate and Severe Anaemia in a High-HIV and TB-Prevalent Adult Population in the Eastern Cape Province, South Africa"

_ijerph, 2023, doi:10.3390/ijerph20043584_

Round 1
Reviewer 1 Report
In this exhaustive study of anaemia in a hospital based community/communities in South Africa the authors do very detailed analysis. Though the conclusions will not be universally applicable this data is quite useful to researchers in the filed of anaemia.
Author Response
Reviewer 1
Comment: English language and style: Moderate English changes required
Response: No specific revisions were requested. We are happy with the use of English in the manuscript. The journal editors can decide if the English needs tweaking.
Reviewer 2 Report
Authors of this manuscript have done a good job in highlighting an important topic on anaemia in the setting of HIV and TB. The use of biochemical testing to determine types of anaemia is a major strength of this study.
Apart from minor spell check and grammatical issues, specific issues to be addressed are.
1. Why did the authors added 20% of the calculated sample size to address the anticipated loss to follow up in a cross sectional study? Line 85-87.
2. The discussion of individual variable is fine, however, I would prefer for the discussion section to be revised and shortened to critically discuss the major findings of the study and how they interconnect into a single story.
Author Response
(1) Comment: Why did the authors added 20% of the calculated sample size to address the anticipated loss to follow up in a cross-sectional study? Line 85-87.
Response: Thank you for the comment. Even though this study was a cross-sectional study, it is atypical in that, patients were screened first before being formally admitted into the study. There could then be a loss to follow up between the screening and laboratory testing as they occurred at different time points. However, to avoid confusion we have since edited this statement and removed the word “…lost-to-follow-up…”; the statement now reads: A further 20% (56) of participants was added to the sample to factor in for anticipated loss of participants during the evaluation process, loss of data and data entry errors.
(2) Comment: The discussion of individual variable is fine, however, I would prefer for the discussion section to be revised and shortened to critically discuss the major findings of the study and how they interconnect into a single story.
Response: Thank you for the advice, it is appreciated. We considered feedback from all the reviewers when making the revisions and consider it has improved the readability of the manuscript.
Reviewer 3 Report
Review of IJERPH-2117846
This is a very interesting paper, but some work is needed. Despite the large number of criticisms in this review, I think the paper has a lot of merit and can be easily fixed up.
I’m the statistical reviewer, with a bit of experience in this area, but not at this level of technicality, so I may misunderstand some of the issues. I defer to your medical knowledge, but may ask for explanations. Consider my questions as opportunities to clarify what you are trying to say to the readers.
Consider adding some phrasing about diagnostic methods or screening to the paper title.
I’m an American and leave out the extraneous “a”s in “anemia” and “hemoglobin.” I also use a lot more commas that you do.
GENERAL COMMENTS
G1. All titles and tables should state the location (Eastern Cape Province, SA) and the years covered by the data (2017-2019).
G2. Tables with regression coefficients or p-values should also briefly state (probably in the footnotes) the types of analysis that produced them (i.e., for each comparison, chi-squared, Kruskal-Wallis, whatever), possibly in generalizations. See specific comments for questions about whether statistical analysis based on random selection is even possible in this study.
G3. My work tends to be in a much poorer country, and it would be of great interest to people in such countries if the data used in this paper could be analyzed in ways that explored alternative methods (and costs) of determining the types and causes of moderate and severe anemia. It is also clear that the findings of this paper may apply to similar settings in South Africa, and maybe some other places, but that similar studies in other locations or with other types of patients, even in high HIV- and TB-prevalent populations, may be different.
G4. Is it possible that people can have both IDA and AI?
SPECIFIC COMMENTS
Line comment
S1 82 Given that your sample source is medical facilities, it is unreasonable to think that the prevalence of anemia would be nearly as low as that in the general population. You probably could have had a better preliminary idea by examining a small number of charts (or getting billing data, perhaps). Also, given that you used purposive sampling (line 89), by which I believe you mean that you looked for the people most likely to be anemic, the assumption of a low rate is ridiculous. Also, please explain briefly the modalities of this purposiveness. Did the nurses look for the sickest people? What kinds of suggestions/directions were they given? Purposive sampling is likely to have induced various biases in the data that may affect the questions you are asking, which I take to be (1) what tests are best at distinguishing between IDA and AI? (2) among patients with moderate or severe anemia, what proportion have IDA and what proportion have AI? (3) which diagnoses are most likely to be associated with severe anemia? To these I add, (4) what are efficient ways of distinguishing between IDA and AI in low resource settings? (5) for the common diagnoses, what are the proportions of IDA and AI? And (6) do the proportions of IDA and AI vary by facility (or facility type)? This would be an indication that the diagnostic algorithm could or should vary depending on location. Obviously, it is up to you whether you think any of my questions are worth adding to your paper.
S2 Using the single proportion equation to determine the sample size is NOT appropriate. Your title says you are interested in causes (PLURAL), thus implying at least a double proportion formula. And, indeed, you end up with multiple logistic regression. Using a more realistic (e.g., .2) value for p and still requiring tight confidence intervals would have led to a sample size of 1600 even using this equation, and using a more pessimistic (in the sense that it requires the largest sample) .5 would require 2500. You are lucky that your sample prevalence was much higher than .03, since 3% of 335 subjects would have been only 10 with moderate or severe anemia, and you would not have been able to do any subanalysis, even as in Table 2 or Figure 1.
Nonetheless, you have the sample you have, and you should recast this paragraph as something like: we made a mistake in our sample size calculation. Given the number we got, we had power=.8 to find differences in proportions of aa vs. bb, or cc vs. dd (give a couple of extremes that make sense) or odds ratios of at least zz.
S3 The rest of the paper (and this review) treat the people found to be anemic as a reasonable sample of anemic people. Well, not always…
S4 How many people were approached to get the initial sample of 1327? How many of the PLWHIV positive patients were not taking ART? The breakdown I really want is (1) not previously diagnosed; (2) previously diagnosed but not currently taking ART; (3) taking ART. Based on the low CD4 counts and more than half the PLWHIV having non-suppressed viral loads, I gather that group 3 was a minority.
S5 102-103 Using the MDRD-CKD equation without the “race factor” is not the equation one would get if one estimated from the whole population ignoring “race.” However, I realize that what you did is commonly done these days.
S6 142 Median(IQR). Although we say “IQR,” which you define properly, the whole point of giving these values, rather than the mean and standard deviation, is that the distribution is not likely to be symmetric, and the quartiles are probably not equidistant from the median. You may use whatever terminology you like (e.g., median (quartile bounds), median (IQR)), but please show Median (Q1, Q3).
S7 142-144 You do not mention Fisher’s exact test, though it appears in table footnotes.
S8 148 “purposeful selection of variables” I take this to mean some sort of “hand selection,” as opposed to automatic selection (forward, backward, stepwise), perhaps forcing in variables of interest to the investigators and ignoring variables that were not of interest (?age?).
S9 Table 1 The footnote says 3 asterisks for FET, but I don’t see 3 asterisks in the table. Perhaps a remnant from a “former life.” Given the small numbers in the 18-19 age group, you should consider merging this group with 20-29 for analytic purposes.
S10 Are the 110 patients with missing TB data the same as those not analyzed because they had probable TB based on x-ray? This large number is probably worth a footnote.
S11 MRH, being a higher level facility, obviously has a sicker patient population (or so it seems, given that no real conclusions can be drawn in the face of purposive sampling).
S12 Table 2 Since your “real” definition of anemia is based on lab testing, perhaps change the title of this table to “Clinical conditions of participants with hemoglobin < 10 by HemoCue, ECP, SA, 2017-2019.”
S13 Based on this table, and especially after you reduce to the group of moderate and severe anemia by lab test, the only diagnoses with sufficient numbers to even think about analyzing are HIV, HTN, TB, and DM (type I or II?). CKD is marginal. Nothing serious can be said about cancer and heart failure.
S14 Figure 1 Given the purposive sampling, it is not even clear that the relative proportions of IDA and AI are stable, or that any “shortcuts” that might be based on eliminating small minority subsets (e.g. 232 of 288 with ferritin over 30 and inflammation had AI), though disregarding TSAT is obviously bad for the 34 patients who would be misdiagnosed without it.
S15 193-198 I suppose that this is useful information for clinicians and researchers.
S16 199-200 Specificity of 89% is not that much better than simply saying “no” for all (specificity of 80%).
S17 Table 4 Why is the left part of the “yes” line bolded?
S18 Fig.1, T2-5 This section is where it might be useful to make comparisons within the larger diagnoses. I defer to your medical knowledge.
S19 218-231 I do not understand the purpose of these detailed reports on small numbers of patients (unless much larger fractions have been reported elsewhere), but I defer to your superior medical knowledge.
S20 233-236 How many patients with macrocytosis had HIV disease (whether or not you have drug history)?
S21 241-246 I had a lot of trouble reading this paragraph, especially figuring out what the numerators and denominators represented. I think I finally figured out that of the 219 patients with HIV, HIV was the single cause of anemia in only 136 (<2/3). I assume that “HIV” as a cause means consequences of HIV and its treatment. You can’t count HIV and lamivudine separately. I guess I find this an uncongenial way of thinking about these numbers. Also, is it of any interest to distinguish between IDA and AI, or is it clear that all (or almost all) the PLWHIV have AI? Since this is a large group, I suppose that figuring out what’s really going on has relevance to the widespread prescribing of iron (with or without folate) to PLWHIV, for example.
S22 Table 6 Please add “among patients with moderate or severe anemia” (or the equivalent) to the title. This would obviate the need for “(severe vs moderate anemia)” in the variable column header. I am not sure that, given all the issues of sampling, this analysis is particularly valuable. If you think it is, you should probably explore interactions. For example, you might consider making a variable of HIV*TB with 4 categories and using “neither” as the reference category. Clearly, this is only reasonable for predictors with large-ish numerators and denominators in column 2. Would the HemoCue result be of any interest as a predictor here? Since this table does not distinguish between IDA and AI, why not use the largest group for which you have all these variables (which is, I think, all those with laboratory hemoglobin testing)?
S23 And please remember that these are Odds Ratios, not Risk Ratios, so when they are large, you can’t necessarily say that group 1 is <OR> times as likely as group 2 to have the outcome. Also, in the multivariate context, A|B and B|A are not equivalent. And why would I want to think in terms of people with severe anemia being 3 times as likely to to have TB than people with moderate anemia. You aren’t going to use severe anemia as a reason for a TB work-up.
S24 Finally, given purposive sampling and the use of a subsample, it’s not clear that the p-values are meaningful. You should probably delete them or add a sentence in the Methods section saying that they are “purely descriptive.” I really don’t think you should talk about “statistical significance” here or elsewhere in the paper (e.g., line 277).
S25 Assuming that you choose to keep this model and this table, how did you deal with missing data? This information belongs in the Methods section.
S26 Table 7 This table is really hard to read and might well be two tables, one for HIV and one for TB. Here again, the HIV-TB overlap complicates things. Obviously, for some of the measured included, the type of anemia is relevant.
S27 292 The age group 20-24 has not previously been mentioned in this study.
S28 299 Patients are likely to be sicker than general community members.
S29 334 The CI as written has (higher value, lower value) AND does NOT include the point estimate. Perhaps there is a typo.
S30 349-350 Please clarify that this is talking about the subset of those with moderate or severe anemia. The “3 times” issue again.
S31 377-379 Are you saying that the general population worldwide has more CKD than Africans? If so, this could reflect the age distribution, less complete case ascertainment in Africa, or differences in prevalence of risk factors for CKD, such as better survival of diabetes patients in richer countries (but a lot of the rest of the world is poor)?
S32 381 Even though you are not using the “race factor?” Personally, I have never seen any estimates of R squared for any of the eGFR equations, with or without the “race factor,” and I suspect that they are not great.
S33 390 Should you say “elsewhere in South Africa” or “in a representative sample of the South African population?”
S34 404 Should “deficiency” come after “folate?”
S35 415-417 A simple table in the Results section or in Supplementary materials would be helpful.
S36 432 Did you say this in the Methods section?
S37 435-439 I think the order should be (1) HIV, ID, TB the most common causes; (2) AI the most common type; (3) greater likelihood of TB among severe than among moderate (and you have to make this comparison explicit). This last is probably true, even if the “3” is wrong. When you say it this way, it should be based on raw numbers, not a model.
==================================================================
My take-home messages:
Using shortcuts or less distinguishing biochemical markers in diagnosing anaemia as IDA or AI may leave a significant fraction of patients misdiagnosed. It is not clear to me what would have happened if the first cut had been based on inflammation, e.g., what fraction of patients with ferritin under 100 have inflammation (current unknown for the 26 with ferritin under 30, but that’s a small number, under 1/3 of those with IDA).
More than half the patients with moderate or severe anemia had more than one cause (implying that if one stopped too soon in the work-up and treated the first cause found, the patient would not be likely to improve). Only among the PLWHIV was HIV the sole cause in more than half the cases.
==============================================================
Another question: Your models include sex but not age. Is age not a good predictor? (or, possibly age*sex?)
Reviewer 4 Report
the authors have written a well-structurated work without significant shortcomings so as to be accepted in the present form.
The work has no particular shortcomings, however in our opinion it does not arouse great clinical interest.It might be more interesting, for example, to make some clarifications. 1) By tuberculosis the authors mean pulmonary or extrapulmonary tuberculosis or both?. 2) Which CDC class do HIV+ patients belong to? Is viral replication suppressed or not? What is the CD4 value and above all what is the CD4 nadir? How long is the duration of HIV infection? Is the viraemia suppressed? If viraemia is suppressed, is it stably? Did the HIV+ patients have opportunistic infections? Best regards
Round 2
Reviewer 3 Report
Review of IJERPH-2117846 v2 Anaemia in South Africa
General comment: Please add ‘2017-2019” to the title and location and years to the titles (or at least footnotes) of tables and the legends of figures.
Note that the number of comments has vastly decreased (so be happy).
Specific comments
Line comment
S1 80-85 I understand that this is what you did, and that you discuss this as a weakness in the Discussion section. Here I’m trying to do a little education. There are two problems here: (1) you vastly underestimated the prevalence of anemia (at least, as I read this section, you expected it to be 3%, and you were looking for a 2% margin of error. If you mean “approximately 2%,” “+/-2% is not a good way to express it in formal writing), and that is an even worse underestimate; (2) you used the single proportion formula. As a result of expecting a low prevalence, you computed a relatively small sample size. If the prevalence had really been 3%, even on the original 1327 people, you would have had about 40 subjects with moderate-severe anemia, hardly enough to do any multivariate analysis on (or even to report in a convincing way that most of them had AI). Also, I do not know how you could think this was a plausible estimate, given the South African data you cite in the Discussion. Fortunately for you, the prevalence was much higher and you can actually do some statistics. Next time, try to be more realistic in your power calculation, and if you plan to do any comparisons (i.e., anything interesting beyond just stating a prevalence), use a more appropriate sample size calculation, probably more sophisticated than the double proportion formula.
S2 Table1 You grouped 18-19 with 20-29, as I suggested, but did not change the row stub “20-29,” which should now be “18-29.”
S3 Table 2 and general: I am still somewha t confused about what fraction of the PLWHIV had already been diagnosed and were taking ART. Based on "Suppression” it looks as if 180 subects (of 471) had previously diagnosed HIV. But 231 have CD4 measurements (? Including newly diagnosed?). This also is an issue in Table 7 regarding VL and CD4 in the TB columns. Given that you have apparently substantial numbers in these two categories of PLWHIV, you probably have bimodal distributions of VL and CD4 (and, for all I know, categories of anemia). It might be interesting to consider these two subsets independently as predictors of severe anemia.
S4 198-199 I object to using the word “prevalence” to describe a proportion in a subset determined by something that’s not an obvious demographic (age, sex, location) or practical (attending clinic, hospitalized) group, such as people who were moderately or severely anemic by lab Hb test. “Proportion” is better. Given the selective nature of this whole study, “prevalence” is probably a word that’s better not to use.
S5 279 Please change “unit” to “g/dL”
S6 Table 6 What is the number of complete cases used in the multivariable analysis?
S7 279-280 What is a unit of severity? This is a nonquantitative class. Probably better to write that an increase of1 in RPI increases the odds of severe anemia(given that the patient has either moderate or severe anemia) by exp(1.7) (=5.5) and a 1 g/dL increase in MCHC increases the odds of having severe anemia (given that the patient has either moderate or severe anemia) by exp(.5) (=65%). Just in case this is not clear to you, the results of a logistic analysis should be reported as something like “an increase of x (you can use more than 1 as the interesting increment) in <predictor variable> gives a xx% increase/decrease in the odds of <the outcome>.”
S8 Several of the variables listed in this tables, but in particular MCHC and RPI, have not yet appeared in tables. It is desirable to show the distribution of the predictor variables in the basic characteristics tables.
S9 Table 7 I did a simple “analysis” of drawing in arrows up or down for all cases of p<.05. The arrows are in the same directions across the 4 subgroups of the table (where relevant). This means that differences between disease positive and disease negative go in the same direction within categories of anemia severity. I guess this consistency makes me pretty happy.
S10 286-288 This is badly phrased. Change “were likely to have” to “had.” As a general comment, medians and means are/were higher/lower. There is just one number per group. People within these groups may be more likely to have higher/lower values.
S11 324-325 What does “in settings without patient diagnoses” mean? Diagnoses not made (e.g., population survey)? Diagnoses not reported?
S12 The Discussion section has a lot of comparisons of the form x% of subjects in <group A> had <outcome Y>, compared to y% of subjects with <outcome Y> who were in <group A> in some other study. These numbers are not directly comparable and depend on the distribution of characteristics in the other study (so you can maybe do a “reverse calculation” to get the comparable number. If there are enough data reported in the papers referred to so that you can get comparable values, please do so.
S13 369 Please add “reported” before “OIs,” and possibly before “incidences” in lines 371-2 (if that is what you mean). Is this true (low incidence of PJP and toxoplasmosis) in well-followed African populations?
S14 383 I suggest changing “that increases” to “, increasing” (my only suggestion of a comma).
S15 401-403 Is this because diagnoses are not made promptly?
S16 457-458 Are you just saying that cross-sectional designs are good for estimating prevalences?
S17 466-467 If you had used the same procedures to get a larger sample, you would have ended up with roughly the same prevalence. See my comment above at lines 80-85.
I think that S2, S5, S6, S7, S12 are required before this paper can be published.
